# Biological batch normalisation: How intrinsic plasticity improves learning in deep neural networks

**Nolan Peter Shaw**[ID]*, **Tyler Jackson, Jeff Orchard**[ID]

David R. Cheriton School of Computer Science, University of Waterloo, Waterloo, ON, Canada

* nolan.shaw@uwaterloo.ca

## Abstract

In this work, we present a local intrinsic rule that we developed, dubbed IP, inspired by the Infomax rule. Like Infomax, this rule works by controlling the gain and bias of a neuron to regulate its rate of fire. We discuss the biological plausibility of the IP rule and compare it to batch normalisation. We demonstrate that the IP rule improves learning in deep networks, and provides networks with considerable robustness to increases in synaptic learning rates. We also sample the error gradients during learning and show that the IP rule substantially increases the size of the gradients over the course of learning. This suggests that the IP rule solves the vanishing gradient problem. Supplementary analysis is provided to derive the equilibrium solutions that the neuronal gain and bias converge to using our IP rule. An analysis demonstrates that the IP rule results in neuronal information potential similar to that of Infomax, when tested on a fixed input distribution. We also show that batch normalisation also improves information potential, suggesting that this may be a cause for the efficacy of batch normalisation—an open problem at the time of this writing.

## Introduction

The study of how neural learning occurs in the biological brain has led to the development of artificial neural networks (ANNs) in computer science. The resulting research has largely focused on the ability for networks to learn through altering the strength of their synapses. This learning mechanism has a variety of different forms, such as Hebbian learning [1] and its variants [2], as well as error-based learning, particularly backpropagation [3]. We refer to this family of mechanisms generally as *synaptic plasticity*.

However, there are other biological mechanisms in neural networks that remain far less studied in their artificial counterparts. In this paper, we aim to extend research on one such mechanism: *intrinsic plasticity*. Intrinsic plasticity refers to the phenomenon of neurons regulating their firing rate in response to changes in the distribution of their stimuli [4]. This mechanism seems to have two primary benefits for neural networks. The first is that it controls the energy consumption of a neuron; a biological neuron that is firing all the time consumes far more Calories than a neuron with a low firing rate. This advantage is irrelevant in

**Data Availability Statement:** The data underlying the results presented in the study are third party sources and available from http://yann.lecun.com/exdb/mnist/ and https://www.cs.toronto.edu/~kriz/cifar.html.

**Funding:** The authors received no specific funding for this work.

**Competing interests:** The authors have declared that no competing interests exist.

ANNs, where the cost of storing different firing rates is constant. This may be why machine learning researchers have largely focused on synaptic mechanisms rather than intrinsic ones.

The second benefit is computational and founded in information theory [5]. A neuron that never fires cannot propagate a signal, but a neuron that fires all the time also fails to propagate any information about its inputs. Researchers have built single-neuron models demonstrating that neurons can effectively learn the ideal activation function that maximises the information potential of a neuron for a fixed mean firing rate [6, 7]. As previously noted, artificial networks are unconcerned with maintaining a low firing rate, and so a rule can be used that strictly maximises information potential. Bell and Sejnowski's Infomax rule [8] does exactly this.

Previously, Li and Li used synaptic, error-based algorithms to update the connection weights in ANNs in combination with local, intrinsic rules that maximise the information potential of each neuron by tuning its activation function [9]. This method, which they referred to as "synergistic learning" has demonstrated that an intrinsic update rule improves performance when used in conjunction with the error-entropy minimisation (MEE) algorithm, but their study was limited to networks with only one hidden layer. Recently, deep learning has shown that networks with more hidden layers can have superior performance across domains [10]. For this reason, it remains an important open area of research to evaluate the impact of intrinsic plasticity on learning in deep network architectures.

In this paper, we present a novel intrinsic rule, inspired by—but distinct from—the Infomax rule. We demonstrate that this rule improves learning when combined with an error propagating algorithm for learning weights for shallow networks, mirroring the results of Li and Li for the MEE [11] algorithm. This shows that the benefits of intrinsic plasticity extend beyond the MEE algorithm, and can be implemented with vanilla backpropagation.

We then test the impact of our intrinsic rule, which we dub "IP", on learning in deep neural networks. Our results show that IP improves learning in deep networks and that it makes networks more robust to increases in synaptic learning rates. This indicates that our IP rule may solve the vanishing gradient problem through its influence on synaptic weight updates. To test this, we sample the gradients of a hidden layer during learning. Our results show that the size of the gradients are indeed larger than networks that lack the IP rule.

We also compare our IP method to the related method of batch normalisation (BN) [12], showing that both rules have the same family of solutions. Unlike batch normalisation, our rule is biologically plausible, as it does not require a neuron to look ahead in time to adjust its activation function, nor does it perfectly shift its activation function for every distribution of inputs. We build an incremental version of batch normalisation that is biologically plausible and compare it to our IP rule. Our results suggest that batch normalisation may work because of its impact on the information potential of neurons, rather than the previously proposed reasons of reducing internal co-variate shift [12], smoothing the loss function [13], or performing length-direction decoupling [14].

Finally, we provide some analysis of the IP rule to develop a theoretical explanation for its effect on learning in deep networks and why it solves the vanishing gradient problem. First, we show that a side effect of maximising the information potential is that a neuron's activation function becomes centered over the median of its input distribution. This provides a theoretical explanation for why information maximisation increases the size of activation gradients during learning. We also test the information potential of neurons that use the IP update rule and show that they converge to levels of information potential similar to that of the Infomax rule. Finally, we provide a justification for our choice of update rules.

## Model design and implementation

We start by describing the design of our main contribution: the intrinsic plasticity mechanism. We highlight how our implementation is different from the local, Infomax rule used in [8] and provide some intuition for the changes. These changes are discussed in more detail in the Analysis section. Then, we outline the full algorithm, when implemented in conjunction with a synaptic learning rule. We then present the features of our rule that make it biologically plausible. Finally, we discuss the specific implementation used in our experiments, as well as the datasets used for testing our model.

### Intrinsic plasticity mechanisms

Our IP rule is implemented by taking the input into a neuron, $x$, and applying an affine transformation

$$u = \frac{x - b}{a}.$$

The non-linearity is then applied in the form of the tanh function, We use the tanh function, but other activation functions may be used with different update rules,

$$y = \tanh(u).$$

At the end of each feedforward pass, $a$ and $b$ are updated using the equations

$$a = (1 - \eta)a + \eta \cdot (2\mathbb{E}[\mathbf{xy}]) \tag{1}$$

$$b = b + \eta \cdot (4\mathbb{E}[\mathbf{xy}]\mathbb{E}[\mathbf{y}]) \tag{2}$$

where $\mathbb{E}[\cdot]$ refers to expected value, $\mathbb{E}[\mathbf{xy}]$ and $\mathbb{E}[\mathbf{y}]$ are both computed using the input and output statistics for the current batch, and $\eta$ is the intrinsic learning rate.

The intuition for this rule's behaviour is that $a$ and $b$ scale and shift (respectively) the activation function so that it is centered over the median of the distribution, with the steepness of the sigmoid being adjusted so that it is steeper for narrower distributions and shallower for wider distributions. This is done to adjust the output distribution of the neurons so that it is as close to uniform as possible, since the uniform distribution has the highest information entropy. It also has the benefit of increasing the size of the gradient of a neuron's activity with respect to its input. Visualisations of this behaviour can be found in both [8] and [9], and the effect is illustrated simply in Fig 1.

$$\mathbb{E}[\sigma'(\mathbf{X})] \approx 0 \quad \mathbb{E}[\sigma'(\mathbf{X})] \gg 0$$

This rule differs from the Infomax rule, which applies the transformation

$$u = \alpha \cdot x + \beta, \tag{3}$$

and whose update rules are

$$\alpha = \alpha + \eta\left(\frac{1}{\alpha} - 2\mathbb{E}[\mathbf{xy}]\right) \tag{4}$$

$$\beta = \beta + \eta(-2\mathbb{E}[\mathbf{y}]). \tag{5}$$

Both Infomax and our IP rule have the same purpose—to shift and scale the activation

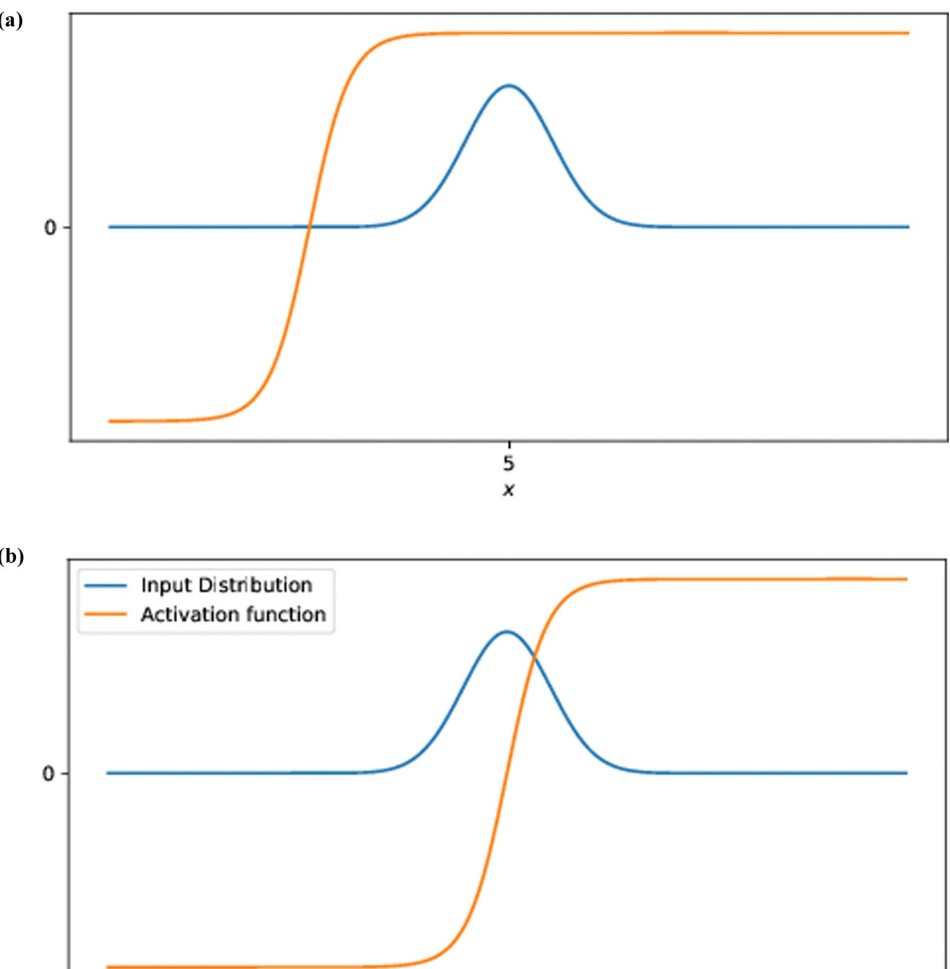

**Fig 1. Effect of the IP rule on the gradient of the activation function.** When the activation function is centered over its input distribution, the gradients of the activation function are much larger. Since error backpropagation uses these gradients as part of a product via the chain rule, centered activation functions propagate larger error gradients than off-center ones.

function such that its output distribution is as close to uniform as possible. We provide additional analysis at the start of the Analysis section showing that our IP rule does, in fact, center the activation function over the median of the input distribution.

## Complete algorithm for a feedforward neural network

Due to the prevalence of machine learning literature, a complete overview of the algorithm used here will omit details that are universal to neural networks. Additionally, unless it is important to specify a particular layer or neuron, subscripts will be dropped to improve readability and no distinction will be made between vectors of dimension one and those of higher dimension.

**Step 1: Initialisation.** The network is constructed with the specified hyperparameters and its weight matrices initialised randomly. The $a$ and $b$ for each neuron are initialised to 1 and 0, respectively.

**Step 2: Feedforward pass.** Inputs $x_0$ are fed into the network's input layer, and the activation function of each neuron is applied

$$u_0 = \frac{x_0 - b_0}{a_0}, \tag{6}$$

$$y_0 = \tanh(u_0). \tag{7}$$

These values are then multiplied by the weight matrix, $\mathbf{W}_0$, with dimensions $m \times n$, where $n$ is the dimension of the input layer and $m$ is the dimension of the subsequent layer. A bias, $B_0$, is added, yielding

$$x_1 = \mathbf{W}_0 \cdot y_0 + B_0. \tag{8}$$

This is then repeated for every layer until the final, output layer, where ReLUs are used in lieu of the tanh function and the IP mechanism is not applied. The output of the network will be referred to as $y_{\text{out}}$.

**Step 3: Update intrinsic parameters.** After the feedforward pass, the intrinsic parameters for each neuron in the network are then updated using the update rules specified above. Note that the IP parameters here are updated every batch, rather than once every epoch (as done by Li and Li).

**Step 4: Compute error and update weights.** The output of the network, $y_{\text{out}}$, and the target output, $t$, are then used to compute the error for some given loss function. Typically, the $L_2$ or cross entropy error are used, however this algorithm is agnostic to the choice of loss function. Since the following experiments will be comparing performance for classification tasks, cross entropy will be used for our purposes. The weights of the network are then updated using the Adam algorithm [15], which is a variant of backprop.

**Step 5: Loop or halt.** This process is then repeated for each batch in the data set. Then, if the halting condition is reached (either by having sufficiently low loss or running for enough epochs), the algorithm terminates. Otherwise, the algorithm re-randomises the data set and returns to Step 2.

## Biological plausibility

The IP rule, as implemented in the above algorithm, possesses many biologically plausible features. First, it is spatially local. The update rules for $a$ and $b$ only require information about the neuron's input and output, $\mathbf{x}$ and $\mathbf{y}$. Second, the rule is temporally local and consistent. The mechanism stores information about the neuron's input and output distributions using persistent parameters, and the statistics used to compute the update rules only observe a relatively small number of samples in the past. It is likely that the brain is capable of computing the described statistics through the regulated, local supply of ambient chemicals.

Furthermore, updates to the intrinsic parameters only occur after a neuron has observed an input. This distinguishes intrinsic plasticity from the batch normalisation (BN) method, which applies a transformation to inputs for the current batch, requiring that a neuron update its activation function prior to actually seeing some inputs. This ability to look forward in time improves the effectiveness of BN, but it is unlikely that this is biologically plausible, unless it is later discovered that dendrites are capable of performing normalisation tasks prior to changes in the somatic membrane potential. Batch normalisation also computes its bias using the input to a neuron, whereas our intrinsic rules use the activity of a neuron to update its activation function. For both of these reasons, the intrinsic rules we present seem to be more biologically plausible than conventional implementations of BN.

While the IP rule possesses these biologically plausible features, we have implemented and tested our intrinsic rule's effect on backpropagation—a learning rule that has been widely criticised for its biological implausibility. However, many papers have suggested that the biological implausibility of backprop is overstated [16]. Also, other algorithms that implement local learning rules have been shown to converge to the error gradients computed during backpropagation [17]. For these reasons, we feel that the use of backpropagation is justified, and may be treated as a simplifying abstraction of other biological mechanisms, as the focus of this work is studying the effect that intrinsic rules have on learning—not the implementation of synaptic rules.

It should be noted that our computational model omits many biological features for the sake of simplicity and more clearly illustrating how the intrinsic rules operate in isolation. For this reason, we do not claim that our algorithm replicates actual brain function. Nevertheless, our rule possesses qualities of spatial and temporal locality that enable its success in both online and distributed settings. New, emerging hardware architectures, such as neuromorphic processors, are much closer to biological brains than traditional computer hardware and would particularly benefit from learning rules that are spatially and temporally local.

## Implementation

The networks used in the following sections were built in Python using the pytorch package. A github repository containing the code used in this project can be found at https://github.com/Shawfest/ip.

## Data sets used

The two data sets used to run the following experiments are the MNIST database of handwritten digits [18], and the CIFAR-10 data set [19, 20], which consists of images from ten classes of objects. Example images for each of these data sets are shown in Fig 2.

## Experiments

In this section, we study the behaviour of the IP rule for both shallow and deep networks. We compare its performance against other established methods, namely: standard vanilla network with backprop, Infomax, and a modified version of batch normalization.

(a)                                                    (b)

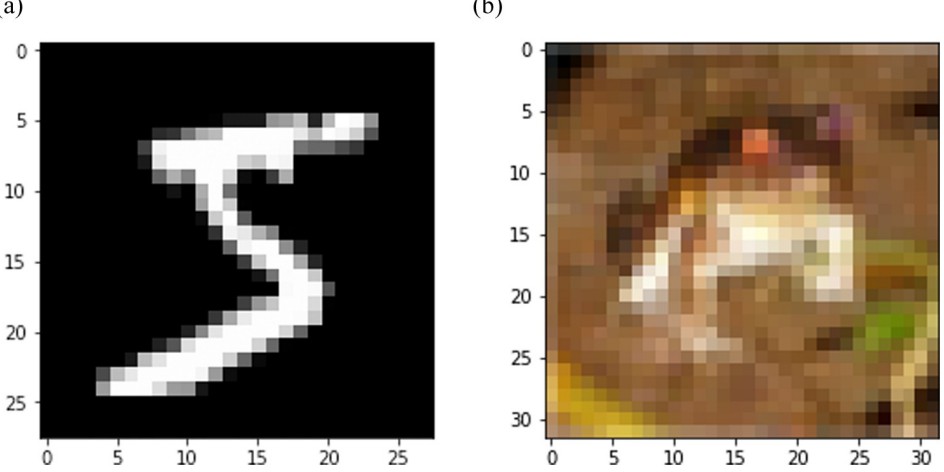

**Fig 2. Example inputs used for experiments.** The above images are two inputs from the MNIST and CIFAR-10 datasets. (**a**) A hand-written five in MNIST. (**b**) A frog in CIFAR-10.

The goal of our experiments was not to design a competitive algorithm, but rather to test the effect on learning of the intrinsic plasticity rules in isolation. For this reason, we chose a basic, fully-connected feedforward architecture for all experiments. In practice, we set the learning rate for the bias, $b$, to be half as large as the gain, $a$.

## Effects of IP when using backpropagation

The easiest hypothesis to test was that the benefits of intrinsic mechanisms were not restricted to the MEE algorithm. Furthermore, since the IP rule differs from that used in [9], testing it on a shallow network to mirror the studies done by Li and Li would help establish a foundation that the IP rule can be beneficial. This test was also done to ensure that IP actually works.

A network with the IP mechanism and a standard network without IP were trained on MNIST. Both networks had fully connected layers, with the input layer having 784 neurons to match the size of the MNIST digits, the hidden layer had 50 neurons, and the output layer had 10 neurons—one for each class of digit. For the sake of fair comparison, the weight matrices for both networks were initialised to the same values. The synaptic learning rate for the Adam algorithm was set to 0.03, and the intrinsic learning rate, $\eta$, was set to 0.0001. This experiment was run ten times for different weight matrix initialisations, with the results averaged and presented in Fig 3.

The same experiment was run for the CIFAR-10 data set with the input layer size changed to 3072 to match the three colour channels of the $32 \times 32$ images. To account for the increased difficulty of the problem, both networks had their synaptic learning rates turned down to 0.001 and were given 150 neurons in their hidden layer. These results can be seen in Fig 4.

The results in these experiments are similar to those found by Li and Li [9]. Networks with IP converge to better solutions than networks without IP. It is worth noting that the improved performance is likely not due to improved gradients, since there is only one hidden layer. Rather, it is likely that the improved performance is due to the improved efficiency of the neurons, as observed in by Li and Li [9]. These results demonstrate that information maximisation is beneficial for a broader range of synaptic learning algorithms than the MEE algorithm.

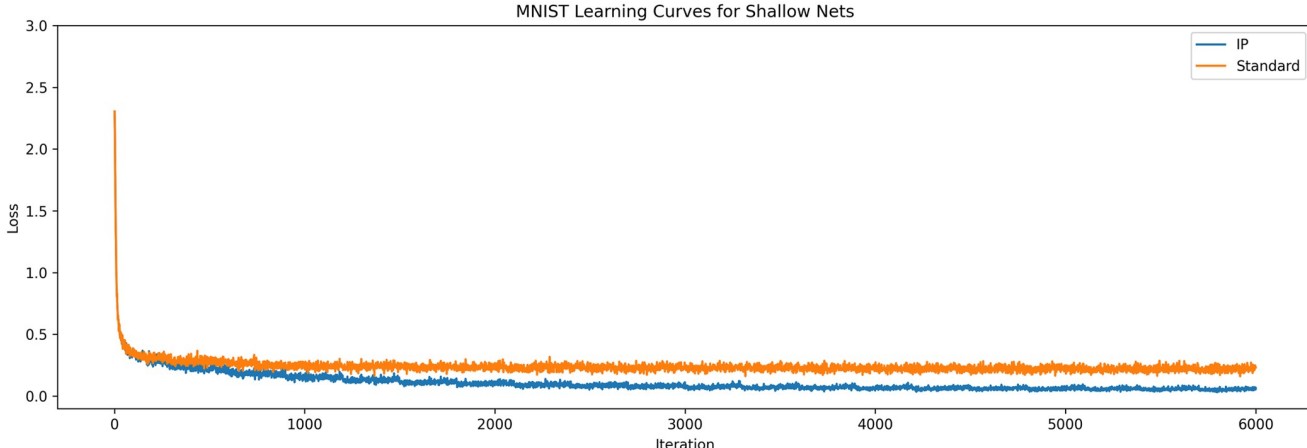

**Fig 3. Learning curves for shallow networks.** The averaged learning curves for both IP and standard networks trained on MNIST across 20 epochs. Observe that the IP networks achieve higher performance (lower loss) after training than their standard counterparts.

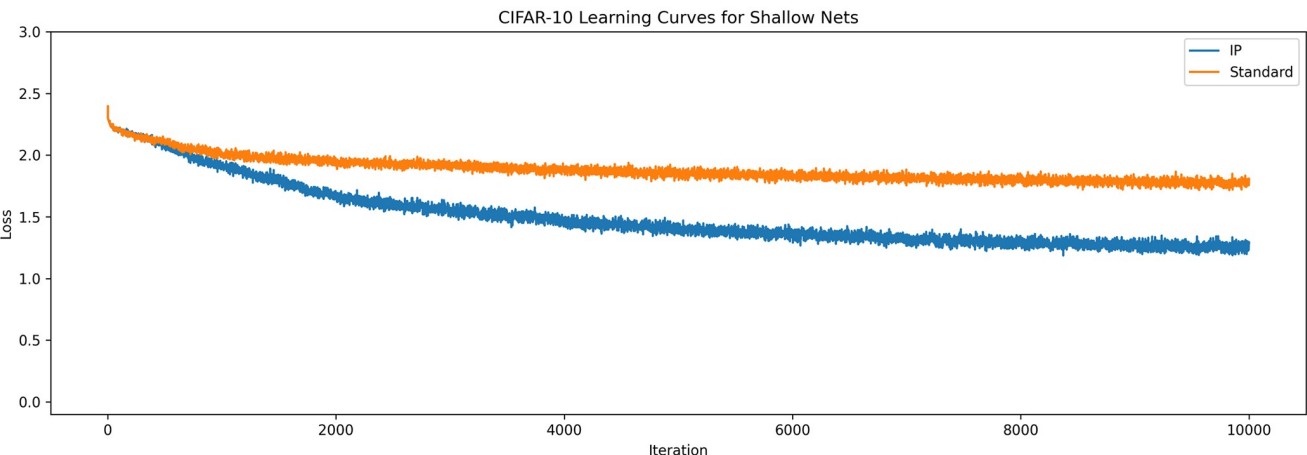

**Fig 4. Learning curves for shallow networks on CIFAR-10.** The averaged learning curves for both IP and standard networks trained on CIFAR-10 across 40 epochs. Again, the IP rule improves upon the performance of a standard network.

## IP improves learning in deep ANNs

Having shown that IP improves learning in shallow networks, we then tested our primary hypothesis: that the IP mechanism improves learning in deep ANNs and is robust to increases in synaptic learning rates.

To test this hypothesis, we designed a series of experiments that would compare IP networks to standard networks for various synaptic learning rates. Inspired by the experiments in [12], we suspected that, while an improvement in learning would be seen for small synaptic learning rates, the benefits of IP would become clearer for large synaptic learning rates. This is because standard networks tend to fail due to divergent behaviour and having activities stuck in the saturated regimes of the activation function.

Like the previous experiments, the networks tested were fully connected but with seven hidden layers rather than one. The results for MNIST are presented in Fig 5. As suspected, IP provides a small improvement in learning when the synaptic learning rate is small, but becomes more pronounced as the synaptic learning rate is increased. A standard network will slowly lose performance and eventually diverge as the synaptic learning rate grows, but a network with IP will continue to learn effectively for a longer period.

The results on CIFAR-10 shown in Fig 6 maintain a similar pattern. For small synaptic learning rates, IP outperforms a standard network by a small margin. However, once this learning rate is turned up, this gap widens considerably and networks with the IP rule begin to perform considerably better than their standard counterparts.

The results for this experiment support the hypothesis that intrinsic plasticity can improve synaptic learning. Networks with IP converged to lower losses than their standard counterparts in all the tests (MNIST seems to be simple enough that almost any deep network will converge to low losses). Finally, Fig 7 shows that the gradients of the activation functions are much larger than those found in standard networks. This was tested by re-running the MNIST experiment with the synaptic learning rate set to 0.005, while we recorded the values of $\frac{\partial y}{\partial u}$ in an intermediate layer of the network. This strongly supports the hypothesis that local, intrinsic rules can help solve the vanishing gradient problem.

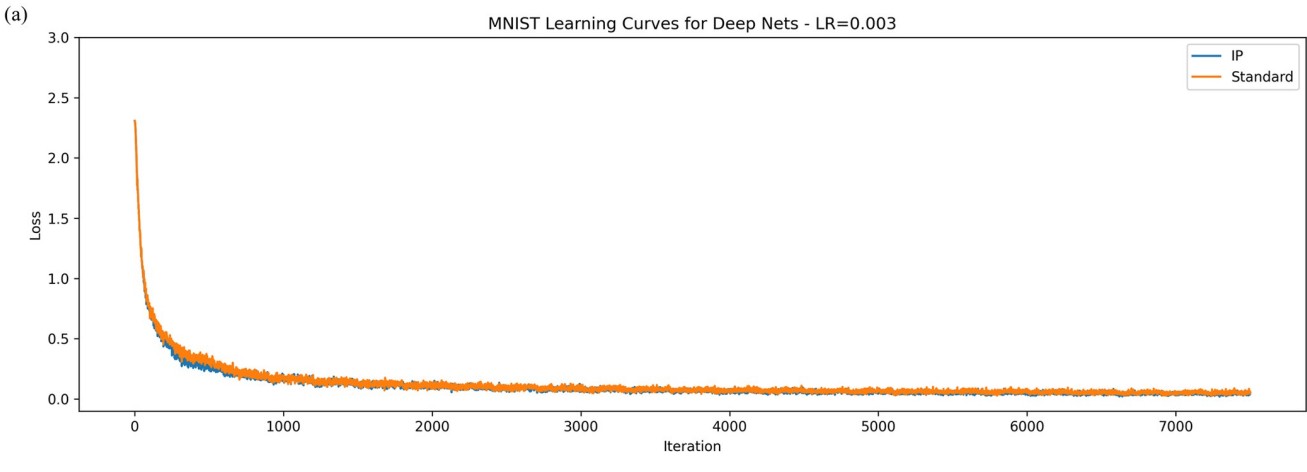

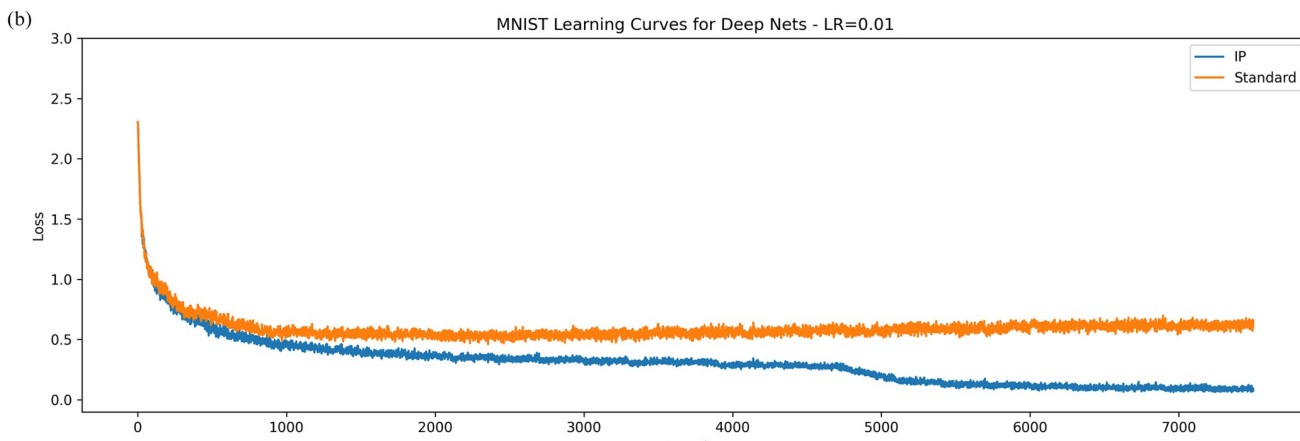

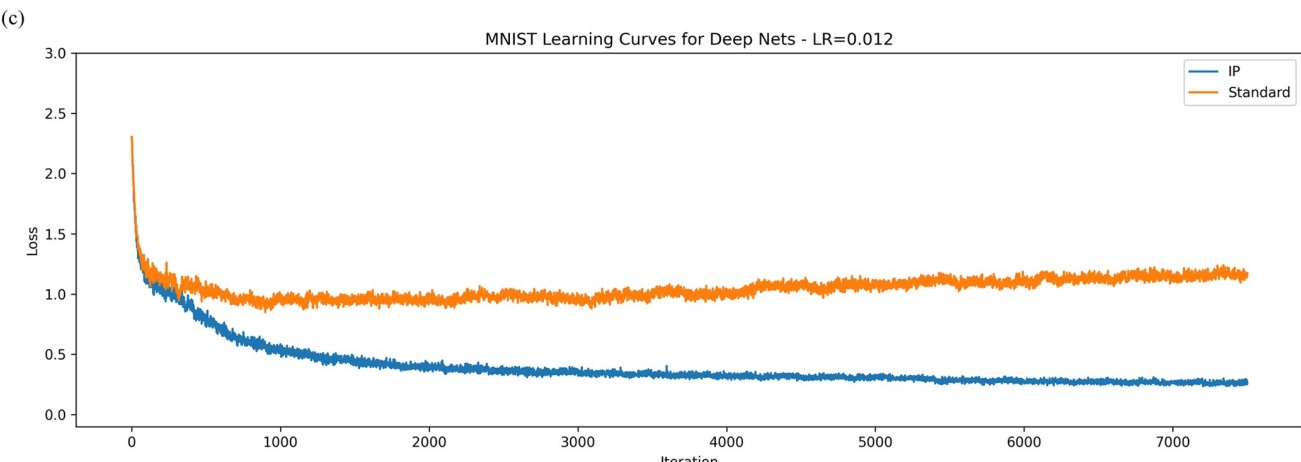

**Fig 5. Learning curves for deep networks on MNIST.** The averaged learning curves for both IP and standard networks trained on MNIST across 20 epochs. The synaptic learning rates for each are, in order, 0.003, 0.01, 0.012.

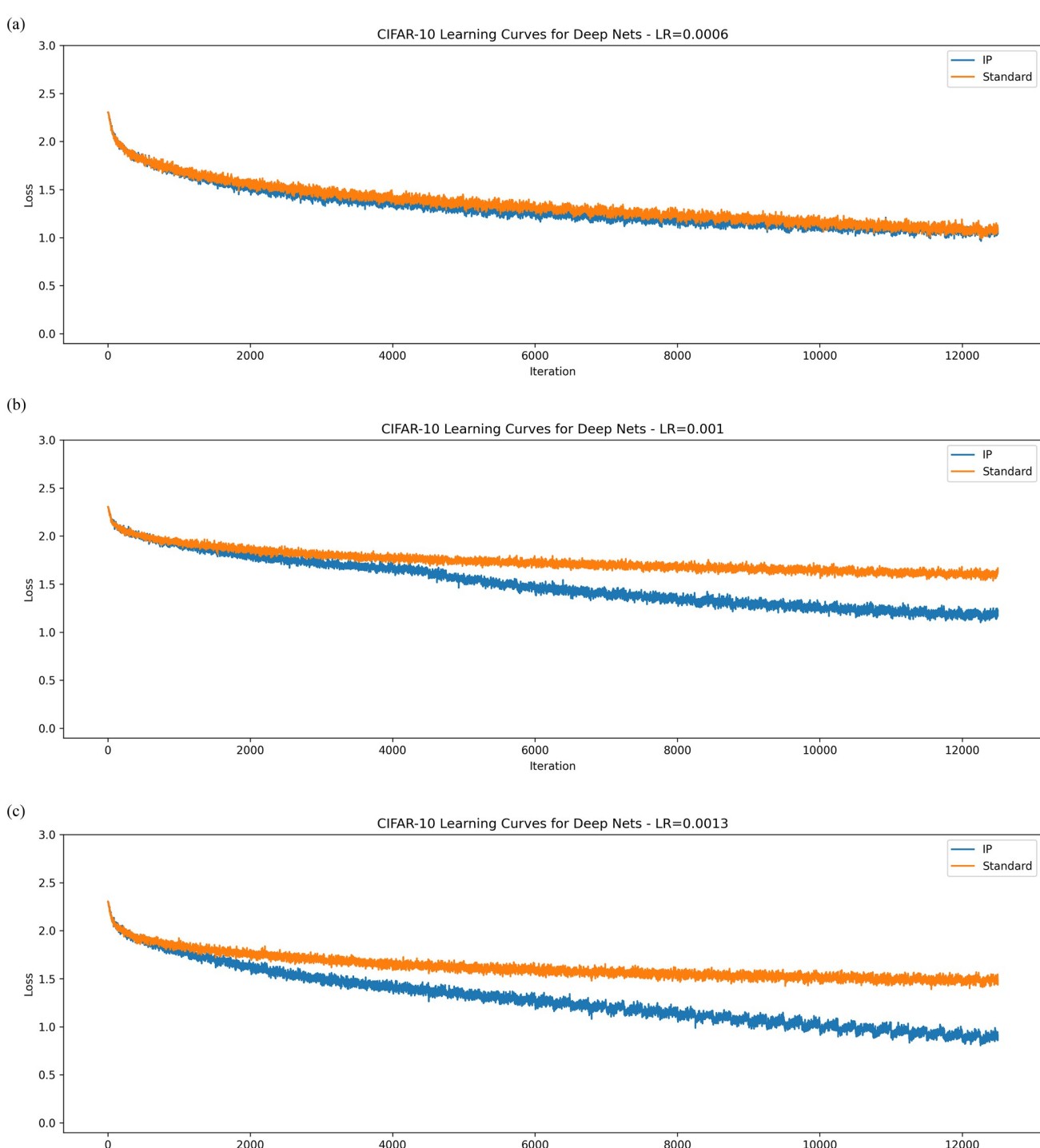

**Fig 6. Learning curves for deep networks on CIFAR-10.** The averaged learning curves for both IP and standard networks trained on CIFAR-10 across 40 epochs. The synaptic learning rates for each are, in order, 0.0006, 0.001, 0.0013.

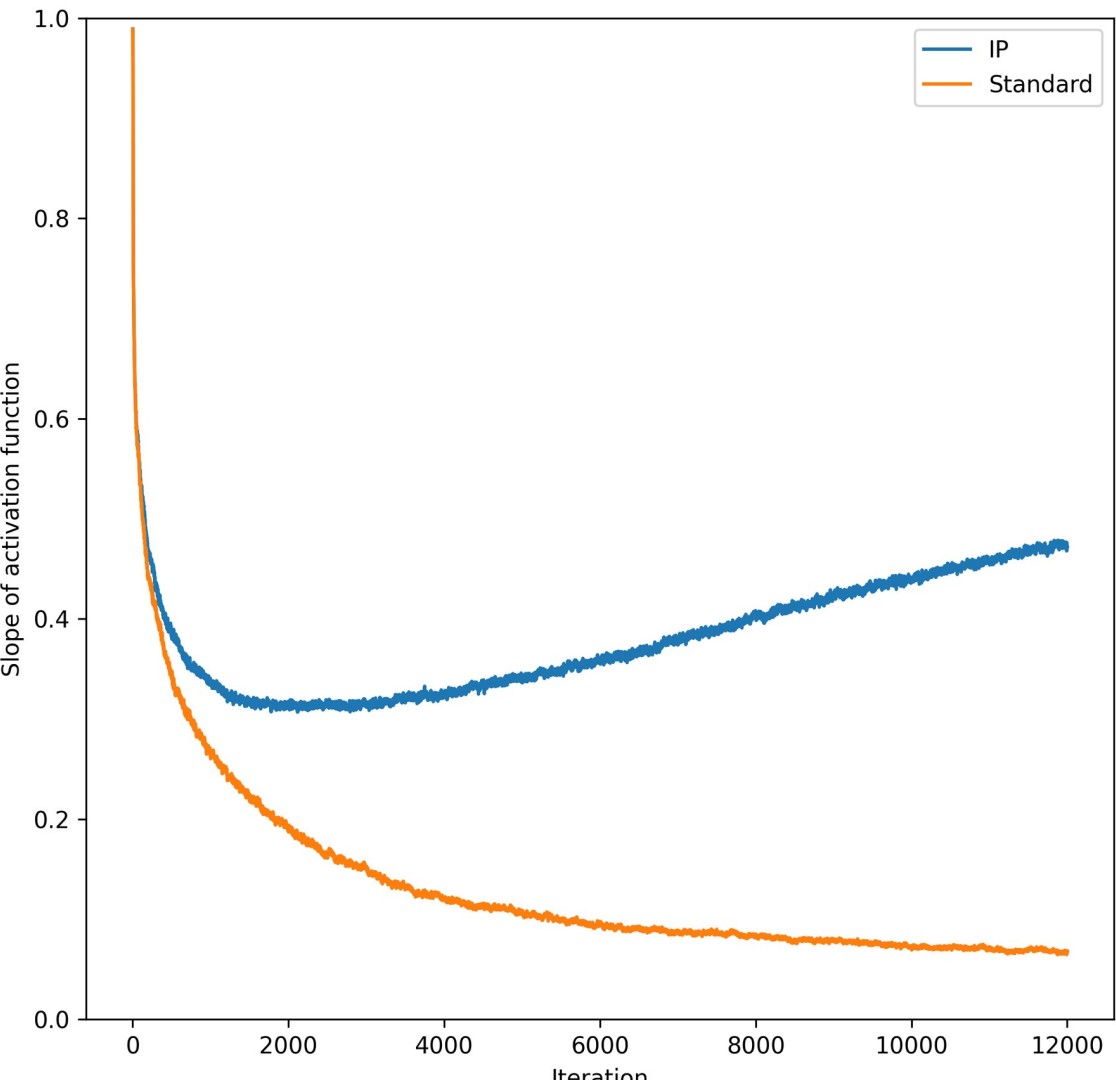

**Fig 7. Value of activation gradients.** The graph shows the average value of $\frac{\partial y}{\partial u}$ for a particular layer during training. The fourth layer of the network (i.e. the third hidden layer), was chosen (the full network had 9 layers in total). As you can see, the gradient of $y$ when IP is implemented is much larger than a standard network over the course of learning.

## Comparing IP to the Infomax rule and batch normalisation

Having demonstrated the effects of IP on deep ANNs, we then compare the IP rule to the original Infomax rule. This was done to test the hypothesis that a more stable version of the Infomax rule could be implemented. Additionally, as stated before, the IP rule bears striking similarity to batch normalisation. Both rules are applied as affine transformations on the input, parameterised by two values. For this reason, both rules are capable of learning the same families of functions. However, unlike IP, batch normalisation works by transforming its input to each hidden layer using the statistics of the current batch, rather than storing this information locally in persistent parameters. For this reason, batch normalisation has the biologically-implausible advantage of allowing nodes to update their gains and biases perfectly and before observing their input.

To conduct a fair comparison of IP to batch normalisation, we implemented BN in an incremental manner like we did with IP. The feedforward phase was the same, but the update rules governing $a$ and $b$ were

$$a_{\text{BN}} = (1 - \eta) \cdot a_{\text{BN}} + \eta \cdot \sqrt{\text{Var}(\mathbf{x})} \tag{9}$$

$$b_{\text{BN}} = (1 - \eta) \cdot b_{\text{BN}} + \eta \cdot \mathbb{E}[\mathbf{x}] \tag{10}$$

where $\sqrt{\text{Var}(\mathbf{x})}$ is the standard deviation of $\mathbf{x}$.

The results of comparing the IP rule to Infomax and BN are shown in Fig 8. The same network hyperparameters as the previous section were used, with the synaptic learning rate set to 0.005 for the MNIST experiment, and 0.001 for the CIFAR-10 experiment.

## Observations

Overall, the empirical results for the IP rule are very compelling. For both MNIST and CIFAR-10, the networks with IP far outperformed networks without. The experiments for shallow networks showed that the local IP learning rule is compatible with synaptic weight updates other than the MEE algorithm. Improvements in performance were even more apparent in deep networks. There it was shown that the gradients of the activation function remain quite large throughout training. This supports the primary hypothesis of this work—that information maximising rules can address the vanishing gradient problem.

## Analysis

### Using neuronal activity to compute the median of the input

The first theoretical observation is that our IP rule biases the activation functions of its neurons such that they are centered over the median of their input distributions, when tanh is used.

**Lemma 1**. *Consider an activation function with input* $\mathbf{x}$ *whose distribution is* $p(\mathbf{x})$. *An equilibrium solution for b occurs when* $\mathbb{E}[\mathbf{y}] = 0$, *where* $y = \tanh(u)$, $u = \frac{x-b}{a}$. *This is a stable equilibrium for b, and is approximately* $\tilde{\mathbf{x}}$—*the median of the input distribution* $\mathbf{x}$.

*Proof*. Since the update rule for $b$ is $b = b + \eta \cdot (4\mathbb{E}[\mathbf{xy}]\mathbb{E}[\mathbf{y}])$, it is clear to see that when $\mathbb{E}[\mathbf{y}] = 0$ is zero, the product, $4\mathbb{E}[\mathbf{xy}]\mathbb{E}[\mathbf{y}]$, is also zero.

Next, we show that $b^* = \tilde{\mathbf{x}}$ is an equilibrium point for $b$. Since tanh is an odd function ranging between $-1$ and $1$, it can be approximated by the indicator function, $1[-1, 1]$, that returns $-1$ if its input is less than 0 and 1 if its input is greater than or equal to 0. The error of this approximation is only large when $u$ is close to zero, which occurs when the function is already close to centered over its input. Hence, for a distribution, $\mathbf{u}$,

$$\mathbb{E}[\mathbf{y}] = \mathbb{E}[\tanh(\mathbf{u})] \tag{11}$$

$$\approx \mathbb{E}[1[-1, 1](\mathbf{u})] \tag{12}$$

$$= -1 \cdot \rho + 1 \cdot (1 - \rho), \quad \text{where } \rho \text{ is the proportion of inputs less than } 0, \tag{13}$$

which equals zero when $\rho = 0.5$. Thus, $b$ continues updating until $\mathbf{u}$ is less than zero half the time and more than zero the other half, i.e. when $b = \tilde{\mathbf{x}}$.

Now, we must show that $\tilde{\mathbf{x}}$ is a stable equilibrium point for $b$. It is sufficient to show that $4\,\mathbb{E}[\mathbf{xy}]\,\mathbb{E}[\mathbf{y}]$ is positive for values of $b$ to the left of $b^* = \tilde{\mathbf{x}}$ and negative for values of $b$ to the right of $b^*$. We do this by showing that $\mathbb{E}[\mathbf{y}]$ is has this property around $b^*$ and that $\mathbb{E}[\mathbf{xy}]$ is always positive in this local neighbourhood.

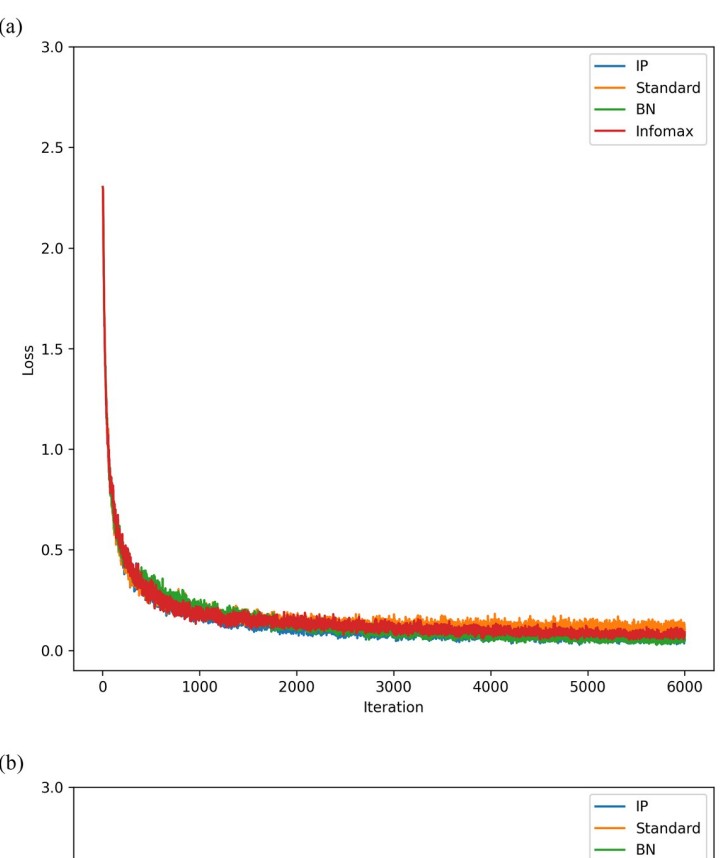

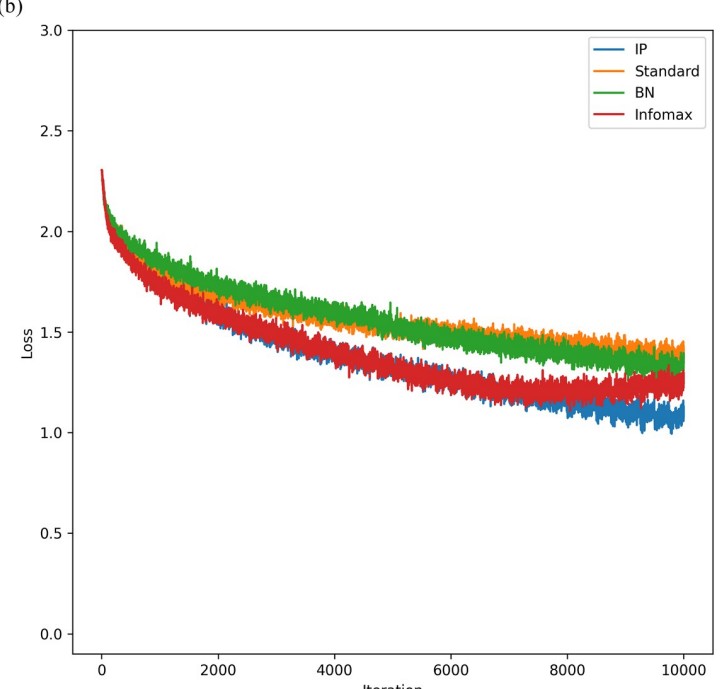

**Fig 8. Learning curves for deep networks using Infomax, IP, and BN.** For this experiment, all three local rules had the same intrinsic learning rate of 0.0001. Again, 10 experiments were done with the results averaged. In both cases, networks that used the IP rule weremore successful than both BN and Infomax. (**a**) MNIST learning curves. (**b**) CIFAR-10 learning curves.

First, consider $\mathbb{E}[\mathbf{y}]$. By the same argument used to show that $b^* = \tilde{\mathbf{x}}$, we can see immediately that larger values of $b$ result in a greater proportion of inputs being below the center of the activation function, so $\mathbb{E}[\mathbf{y}]$ is negative. Mutatis Mutandis, for smaller values of $b$ we have that $\mathbb{E}[\mathbf{y}]$ is positive.

Next, consider $\mathbb{E}[\mathbf{xy}]$ for some fixed $a$ around $b^*$. This is

$$\int_{-\infty}^{\infty} x \cdot \sigma\left(\frac{x - b^*}{a}\right) \cdot p(x) \ dx.$$

Performing the change of variables, $x^* = x - b^*$ such that $p(x) = p(x^* + b^*) = \tilde{p}(x^*)$, gives us

$$\mathbb{E}[\mathbf{xy}] = \int_{-\infty}^{\infty} x^* + b^* \cdot \sigma\left(\frac{x^*}{a}\right) \cdot \tilde{p}(x^*) \ dx^* \tag{14}$$

$$= \int_{-\infty}^{\infty} x^* \cdot \sigma\left(\frac{x^*}{a}\right) \cdot \tilde{p}(x^*) \ dx^* + b^* \int_{-\infty}^{\infty} \sigma\left(\frac{x^*}{a}\right) \cdot \tilde{p}(x^*) \ dx^* \tag{15}$$

$$= \int_{-\infty}^{\infty} x^* \cdot \sigma\left(\frac{x^*}{a}\right) \cdot \tilde{p}(x^*) \ dx^*, \ \text{since } \mathbb{E}[\mathbf{y}] = 0 \tag{16}$$

$$= \int_{-\infty}^{0} x^* \cdot \sigma\left(\frac{x^*}{a}\right) \cdot \tilde{p}(x^*) \ dx^* + \int_{0}^{\infty} x^* \cdot \sigma\left(\frac{x^*}{a}\right) \cdot \tilde{p}(x^*) \ dx^*. \tag{17}$$

Since $\sigma = \tanh$ is odd, we can perform a change of variables to negate $x^*$ in the left-hand term, obtaining

$$\int_{-\infty}^{0} x^* \cdot \sigma\left(\frac{x^*}{a}\right) \cdot \tilde{p}(x^*) \ dx^* + \int_{0}^{\infty} x^* \cdot \sigma\left(\frac{x^*}{a}\right) \cdot \tilde{p}(x^*) \ dx^* \tag{18}$$

$$= \int_{0}^{\infty} -x^* \cdot \sigma\left(\frac{-x^*}{a}\right) \cdot \tilde{p}(-x^*) \ dx^* + \int_{0}^{\infty} x^* \cdot \sigma\left(\frac{x^*}{a}\right) \cdot \tilde{p}(x^*) \ dx^* \tag{19}$$

$$= \int_{0}^{\infty} x^* \cdot \sigma\left(\frac{x^*}{a}\right) \cdot \tilde{p}(-x^*) \ dx^* + \int_{0}^{\infty} x^* \cdot \sigma\left(\frac{x^*}{a}\right) \cdot \tilde{p}(x^*) \ dx^* \tag{20}$$

$$= \int_{0}^{\infty} x^* \cdot \sigma\left(\frac{x^*}{a}\right) \cdot \left(\tilde{p}(x^*) + \tilde{p}(-x^*)\right) \ dx^* \tag{21}$$

Since $x^* \cdot \sigma\left(\frac{x^*}{a}\right)$ is non-negative, and $\tilde{p}$ is non-negative, we have that $\mathbb{E}[\mathbf{xy}]$ is also non-negative. This equals zero precisely when all values for $x^*$ are zero, i.e. when all $x$ take on the same value.

An immediate corollary of this observation is that $\beta = \bar{x}$ (the mean of $\mathbf{x}$) for symmetric distributions. If the input distributions are approximately Gaussian, this indicates that computing the mean, as done in batch normalisation, improves the information potential of a neuron.

## Information potential

The derivation of the Infomax rule is based on the principle of maximizing the mutual information between $x$ and $y = \sigma(\alpha x + \beta)$. In order to accomplish this, the gradient of the mutual information with respect to the parameters $\alpha$ and $\beta$ can be calculated and used to maximize the entropy of the output distribution for any given input distribution of $x$.

Since the IP rule used for this work is derived using simplifying assumptions for the equilibrium solutions of the original Infomax rules, we need to check at least empirically that the entropy of the output distribution is still being increased by this method.

To that end, in Fig 9 we demonstrate that for two different input distributions, uniform of width 4 centered on 1, and Gaussian with $\sigma = 2$ and $\mu = 1$, the IP rule does in fact increase the entropy of the output distribution. For comparison, the effect of the batch normalization rule is also included which also shows an increase in entropy. The entropy of Infomax, which is provably optimal, is also included to provide the theoretical upper bound. Note that the IP rule achieves the same maximum as Infomax (or very nearly) for both distributions. For uniform inputs, the IP rule actually achieves this maximum entropy faster than Infomax. Also note that the batch normalisation method achieves levels of entropy very close to those of IP and Infomax when tested on Gaussian inputs. This suggests that the cause of batch normalisation's success may come from information maximisation, as Gaussian inputs are commonly found in hidden layers during deep learning.

## Choosing the update rules for *a* and *b*

The inspiration for our choice of an update rule for *a* comes from observing that the Infomax update rule for $\alpha$ is

$$\alpha = \alpha + \eta \left( \frac{1}{\alpha} - 2\mathbb{E}[\mathbf{xy}] \right) , \tag{22}$$

so that

$$\Delta\alpha = \frac{1}{\alpha} - 2\mathbb{E}[\mathbf{xy}]. \tag{23}$$

Treating this as a dynamical system and finding the equilibrium by solving $\Delta\alpha = 0$ yields

$$0 = \frac{1}{\alpha} - 2\mathbb{E}[\mathbf{xy}] \tag{24}$$

$$\Rightarrow \quad \frac{1}{\alpha} = 2\mathbb{E}[\mathbf{xy}] \tag{25}$$

$$\Rightarrow \quad \alpha = \frac{1}{2\mathbb{E}[\mathbf{xy}]} \tag{26}$$

Since our transformation is applied as $u = \frac{x-b}{a}$ rather than $u = \alpha \cdot x + \beta$, this indicates that

$$\alpha = 2\mathbb{E}[\mathbf{xy}]$$

should be used. Similarly, for *b* we simply conducted a change of variables from $u = \alpha \cdot x + \beta$ to $u = \frac{x-b}{a}$, yielding $b = b + \eta \cdot 4\, \mathbb{E}[\mathbf{xy}]\, \mathbb{E}[\mathbf{y}]$, instead of $\beta = \beta + \eta \cdot (-2\mathbb{E}[\mathbf{y}])$. To ensure that *a* would properly converge to $2\mathbb{E}[\mathbf{xy}]$, rather than act as an integrator until $\mathbb{E}[\mathbf{xy}] = 0$, the decay term is added, yielding $a = (1 - \eta) \cdot a + \eta \cdot \mathbb{E}[\mathbf{xy}]$.

Note that this ignores the non-linearities from $y = \tanh\left(\frac{x-b}{a}\right)$ in the $\mathbb{E}[\mathbf{xy}]$ term. However, the incremental nature of our rule may lessen the impact of these non-linearities on the value that *a* converges to.

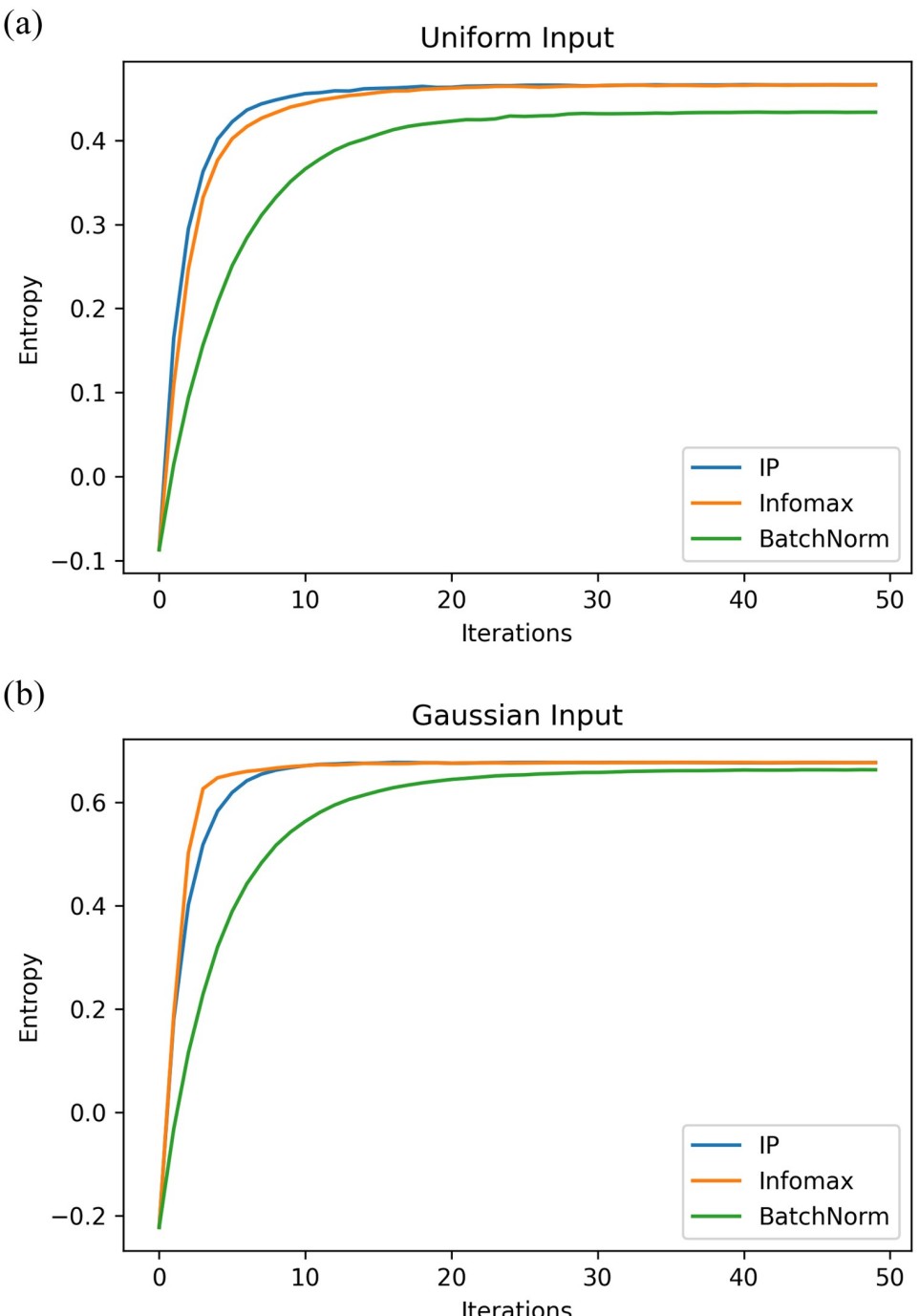

**Fig 9. Neuronal information potential.** To generate these figures, the entropy of the distribution was estimated using the density histograms of the values of y as a Riemann approximation for the integral of the differential entropy. The update rules for each process were applied for multiple iterations on the same batch of 10000 samples. (**a**) Fixed uniforminput distribution. (**b**) Fixed Gaussian input distribution.

## Discussion

### Biologically plausible information maximisation

Studying the computational benefits of neuronal intrinsic plasticity is not new. Triesch first developed a single neuron model of intrinsic plasticity in [6], then improved upon this model in [7] by suggesting a gradient-based rule for maximising the information entropy of a neuron for some fixed mean firing rate. Bell and Sejnowski used an information-maximising rule in [8] and used this for learning the parameters of entire networks to preform signal separation and blind deconvolution.

These rules were implemented in [9] by Li and Li, who combined a local Infomax rule with the error entropy minimisation (MEE) [11] algorithm to improve learning in what they called "synergistic learning". Unlike Li and Li, who only studied the effects of IP on a very small network with one hidden layer, this paper demonstrates the computational benefits that IP confers upon deep neural networks. Also, this paper tests if intrinsic plasticity is compatible with the backpropagation algorithm, which was left unstudied by Li and Li, who instead used it in conjunction with the MEE algorithm [9].

This work also aims to address known stability issues [21, 22] in the Bell and Sejnowski Infomax rule. Unlike Infomax, this work sets $\Delta\alpha$ to zero and rearranges to obtain the solution $\alpha = \frac{1}{2 \cdot \mathbb{E}[\mathbf{xy}]}$. As shown in the experiments, the IP rule does indeed continue to learn as the Infomax rule begins to diverge.

### Explaining batch normalisation

Initially, Ioffe and Szegedy proposed that batch normalisation improved learning in deep networks by reducing what they called internal co-variate shift [12]. More recently, studies have shown that this is not accurate [13, 14]. In [13], Santurkar et al. show that batch normalisation improves performance through improving the smoothness of the optimisation landscape. More precisely, batch normalisation yields smaller changes in loss and the gradients of the loss function are reduced in magnitude. Kohler et al. suggest that batch normalisation creates a length-direction decoupling effect, and that this improves learning [14].

Our work has shown that batch normalisation bears a great deal of similarity to the information-theoretic rules used elsewhere. We have also shown that a modified version of batch normalisation does improve the information potential of a neuron, though not to the levels achieved by Infomax or our IP method. For this reason, we hypothesise that the improvements in accuracy demonstrated by BN may also be related to improved neuronal information potential. Further work is required to prove this or provide more experimental support.

In addition to extending theoretical explanations of why batch normalisation works, this research explains how biological neurons may benefit from employing mechanisms that have a normalising effect on their stimuli. Error propagation on the scale of networks as large as the biological brain may greatly benefit from an intrinsic mechanism that maintains the integrity of both stimuli and error signals.

### Future work

A more in-depth study of why Infomax is less stable can still be performed. Our results only demonstrated that using Infomax as a local learning rule resulted in divergent performance when trained on CIFAR-10. We think the reason that our IP rule might be more stable is that the Infomax differential equation governing $\alpha$ is not Lipschitz continuous, but our differential equation governing $a$ is Lipschitz continuous.

Another direction for future work is evaluating the neuronal information potential in a network during the course of learning. Our work was limited to measuring entropy for fixed input distributions. We are confident that our update rules will exhibit similar levels of improvement in entropy for dynamic distributions.

The goal of this work was to test the effects of IP in a vacuum. For this reason we only compared relative performance on densely-connected, "vanilla" neural networks. Given the breadth of research that has been conducted on various network architectures, testing the impact that IP has on learning in more modern architectures remains an open line of research. Given the locality of our rule, they should be easy to implement in conjunction with other learning models.

Finally, to test the hypothesis that batch normalisation works because it improves information entropy, experimentation should be done that compares the performance of a "hard", or instantaneous, version of IP to conventional batch norm; we have only compared the IP rule to an incremental rule that converges to the parameters used in batch normalisation. Such a rule would be less relevant to biological networks, but may yield benefits to machine learning.

## Conclusion

In this work, we studied the relationship between a local, intrinsic learning mechanism and a synaptic, error-based learning mechanism in ANNs. We developed a local intrinsic rule, dubbed IP (after "intrinsic plasticity"), that was inspired by the Infomax rule. The biological plausibility of this rule was discussed, and it was shown to be more biologically plausible than the functionally similar batch normalisation method.

This work demonstrates that local information maximisation can work in conjunction with synaptic learning rules other than the MEE algorithm, which aims to learn the correct weight updates by minimising the entropy of the error. In shallow networks, the IP rule improves learning on both MNIST and CIFAR-10, compared to standard ANN learning. It was shown that the IP rule improves performance in deep networks while also making them more robust to increases in synaptic learning rates. We also showed that the IP rule increases the average size of the gradients of the activation functions. This supports our hypothesis that intrinsic, information maximisation can solve the vanishing gradient problem. A proof that the IP rule biases its activation function to be centered over the median of the input distribution is also provided to support this claim.

When compared to batch normalisation and Infomax, whose family of solutions were shown to be the same, the IP rule demonstrates an improvement in learning on CIFAR-10. For MNIST, networks with the IP rule do converge to better solutions than BN and Infomax, however the effect is not as pronounced. Furthermore, our IP rule continues to learn on CIFAR-10 when Infomax begins to diverge, suggesting that our rule addresses the instabilities in Infomax that have been previously observed.

Analysis was also provided that demonstrates the IP rule converges to similar levels of neuronal entropy as the Infomax rule, when tested on a fixed input distribution. It was observed that batch normalisation also improves information potential, suggesting that this may be a cause for the efficacy of batch normalisation—an open problem at the time of writing.

## Author Contributions

**Conceptualization:** Nolan Peter Shaw.

**Formal analysis:** Nolan Peter Shaw.

**Investigation:** Nolan Peter Shaw.

**Methodology:** Nolan Peter Shaw.

**Project administration:** Nolan Peter Shaw.

**Software:** Nolan Peter Shaw, Tyler Jackson.

**Supervision:** Jeff Orchard.

**Visualization:** Nolan Peter Shaw, Tyler Jackson.

**Writing – original draft:** Nolan Peter Shaw, Tyler Jackson.

**Writing – review & editing:** Nolan Peter Shaw, Jeff Orchard.

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
