## [Decision Letter · Decision Letter 0]

25 Feb 2020

PONE-D-19-29067

Biological batch normalisation: How intrinsic plasticity improves learning in deep neural networks

PLOS ONE

Dear Mr Shaw,

Thank you for submitting your manuscript to PLOS ONE. After careful consideration, we feel that it has merit but does not fully meet PLOS ONE’s publication criteria as it currently stands. Therefore, we invite you to submit a revised version of the manuscript that addresses the points raised during the review process.

We would appreciate receiving your revised manuscript by Apr 10 2020 11:59PM. To enhance the reproducibility of your results, we recommend that if applicable you deposit your laboratory protocols in protocols.io, where a protocol can be assigned its own identifier (DOI) such that it can be cited independently in the future. For instructions see: http://journals.plos.org/plosone/s/submission-guidelines#loc-laboratory-protocols

We look forward to receiving your revised manuscript.

Kind regards,

Tao Song

Academic Editor

PLOS ONE

Journal Requirements:

2. One of the noted authors is a group or consortium [Neurocognitive Computing Lab]. In addition to naming the author group, please list the individual authors and affiliations within this group in the acknowledgments section of your manuscript. Please also indicate clearly a lead author for this group along with a contact email address.

Reviewers' comments:

Reviewer's Responses to Questions

**Comments to the Author**

1. Is the manuscript technically sound, and do the data support the conclusions?

Reviewer #1: No

Reviewer #2: Yes

2. Has the statistical analysis been performed appropriately and rigorously? 

Reviewer #1: No

Reviewer #2: Yes

3. Have the authors made all data underlying the findings in their manuscript fully available?

Reviewer #1: Yes

Reviewer #2: Yes

4. Is the manuscript presented in an intelligible fashion and written in standard English?

Reviewer #1: Yes

Reviewer #2: Yes

5. Review Comments to the Author

Reviewer #1: The authors investigate two local learning rules applied to shallow and deep neural network learning processes. The learning dynamics is shown to be more stable for a wider range of learning rates. Adding the weight decay rule even makes the network converge to better solutions. These conclusions are supported with two example data sets and some analytical analysis of how the learning rules enhance the learning dynamics.

Although the manuscript contains some insights, the results are mixed and not convincingly presented. Biological plausibility of the local learning rule is a plus, but it should not be the primary emphasis for the additional learning rules, especially since the main learning process is back propagation. Artificial neural networks (ANNs) are at best remotely related to biology. Its main strength is in its performance. As such, biological plausibility added on to ANNs should aim to improve the performance.

The improvements over regular ANNs are seem in the weight decay rule, but the authors also point out theoretical difficulty related to it. The other rule performs even worse than regular ANNs. These make one wonder what we have learned in this work. The manuscript is not well written. At the end the authors even mention “this thesis”, which is a reflection of poor quality control.

The authors might consider considerably improve the manuscript by emphasizing on some variant of the weight decay rule and convincingly make the case that such an addition is useful for stability of learning and also improves accuracy.

Reviewer #2: The paper is well orgnized and easy to read. The intrinsic plasticity is found to be helpful in deep learning. The results are nice, but some related works on intrinsic plasticity in neural networks should be involved, see e.g. [R1-R3]. I recommend to accept this paper once the comments have been addressed.

[R1] Self-orgnized Spiking nueral P Systems

[R2] Axon neural systems

[R3] Spiking neural P systems with plastic synapes

6. PLOS authors have the option to publish the peer review history of their article (what does this mean?). If published, this will include your full peer review and any attached files.

Reviewer #1: No

Reviewer #2: No

---

## [Author Response · Author response to Decision Letter 0]

12 Jun 2020

Editor comments:

The collaboration with the Neurcog lab was removed. This is because we erroneously thought this location was meant to be used to further specify affiliation within our institution. The three authors in the paper are the sole authors of this manuscript.

Reviewer comments:

Reviewer 1:

Comment 1: Although the manuscript contains some insights, the results are mixed and not convincingly presented. Biological plausibility of the local learning rule is a plus, but it should not be the primary emphasis for the additional learning rules, especially since the main learning process is back propagation. Artificial neural networks (ANNs) are at best remotely related to biology. Its main strength is in its performance. As such, biological plausibility added on to ANNs should aim to improve the performance.

Reply to the comment: There are newer recurrent architectures that implement backprop-like learning, such as[Guergiuev et al., 2017, Spratling, 2008, Lillicrap et al., 2016, Bogacz, 2017]. These methods are biologically plausible and have been shown to approximate backpropagation. Hence, they are capable of achieving results comparable to state of the art ANNs. Furthermore, with the advent of neuromorphic hardware approaching, local learning rules that address the vanishing gradient problem are an important direction of research. The biological plausibility of our method is not only important for bridging the gap in theory between machine learning and biology, but also for providing a method that improves learning on hardware that we are currently unable to test on, particularly in an online and distributed environment. We have added to the final paragraph in the biological plausibility subsection that speaks to this importance. Nevertheless, we hope that our revised manuscript will assuage the reviewer’s concern regarding the gap between performance and theory. We’ve identified the error in both methods previously used, and now present a single IP method that is both theoretically sound and demonstrates improved performance across all tests.

Comment 2: The improvements over regular ANNs are seem in the weight decay rule, but the authors also point out theoretical difficulty related to it. The other rule performs even worse than regular ANNs. These make one wonder what we have learned in this work. The manuscript is not well written. At the end the authors even mention “this thesis”, which is a reflection of poor quality control.

Reply to the comment: Good catch. This work does indeed stem from a thesis published in 2019. However, the work has been extended and improved upon.

Comment 3: The authors might consider considerably improve the manuscript by emphasizing on some variant of the weight decay rule and convincingly make the case that such an addition is useful for stability of learning and also improves accuracy.

Reply to the comment: We appreciate the reviewer’s concerns, and agree that the two methods proposed in the first draft were somewhat confusing, and the mixed results failed to demonstrate the value of intrinsic plasticity in a compelling manner. We started this line of research with an intuition that an unintended benefit of information maximisation is its normalising behaviour and that centering neuronal activities over their input distributions could solve the vanishing gradient problem. We wrongly computed the steady-state solutions for the gain, and treated it as an integrator. We have since identified this flaw and the rule presented in this draft is exactly the variant we hope you are looking for. As stated above, it does not have the theoretical flaw of being unable to converge to biases outside of [−1,1] and its performance is in line with what we originally expected to find. In short, the results presented here merge the two methods previously presented and benefit from being both a stronger theoretical foundation as well as improved performance.

Reviewer 2

Comment 1: The paper is well organized and easy to read. The intrinsic plasticity is found to be helpful in deep learning. The results are nice, but some related works on intrinsic plasticity in neural networks should be involved, see e.g. [R1-R3]. I recommend to accept this paper once the comments have been addressed. [R1] Self-orgnized Spiking nueral P Systems [R2] Axon neural systems [R3] Spiking neural P systems with plastic synapes

Reply to the comment: We thank the reviewer for their time and positive comments. We have looked at the suggested papers, and while the work is interesting, we do not feel that they are directly relevant to the topic in this manuscript. More precisely, our impression of these works is that they are primarily concerned with the mechanisms that underlie spike-timing-dependent plasticity and membrane computing can be used to model non-synaptic phenomena in neurons. While membrane computing is expressive enough to describe intrinsic plasticity, the aim of our work is to abstract the various forms of intrinsic plasticity, such as short-term adaptivity and long-term habituation, to focus on how it impacts the dynamics of synaptic updates. We hope that this response is satisfactory. If the reviewer feels that we have missed an important relationship with that work, then we would be happy to address specific comments.

---

## [Decision Letter · Decision Letter 1]

18 Aug 2020

Biological batch normalisation: How intrinsic plasticity improves learning in deep neural networks

PONE-D-19-29067R1

Dear Dr. Shaw,

We’re pleased to inform you that your manuscript has been judged scientifically suitable for publication and will be formally accepted for publication once it meets all outstanding technical requirements.

Kind regards,

Tao Song

Academic Editor

PLOS ONE

Additional Editor Comments (optional):

Reviewers' comments:

Reviewer's Responses to Questions

**Comments to the Author**

1. If the authors have adequately addressed your comments raised in a previous round of review and you feel that this manuscript is now acceptable for publication, you may indicate that here to bypass the “Comments to the Author” section, enter your conflict of interest statement in the “Confidential to Editor” section, and submit your "Accept" recommendation.

Reviewer #1: All comments have been addressed

2. Is the manuscript technically sound, and do the data support the conclusions?

Reviewer #1: Yes

3. Has the statistical analysis been performed appropriately and rigorously? 

Reviewer #1: N/A

4. Have the authors made all data underlying the findings in their manuscript fully available?

Reviewer #1: Yes

5. Is the manuscript presented in an intelligible fashion and written in standard English?

Reviewer #1: Yes

6. Review Comments to the Author

Reviewer #1: (No Response)

7. PLOS authors have the option to publish the peer review history of their article (what does this mean?). If published, this will include your full peer review and any attached files.

Reviewer #1: No

---

## [Editor Report · Acceptance letter]

27 Aug 2020

PONE-D-19-29067R1

Biological batch normalisation: How intrinsic plasticity improves learning in deep neural networks

Dear Dr. Shaw:

I'm pleased to inform you that your manuscript has been deemed suitable for publication in PLOS ONE. Congratulations! Your manuscript is now with our production department.

Kind regards,

on behalf of

Dr. Tao Song

Academic Editor

PLOS ONE